# The Sociodemographic-Professional Profile and Emotional Intelligence in Infant and Primary Education Teachers

**DOI:** 10.3390/ijerph19169882

**Published:** 2022-08-11

**Authors:** Wendy L. Arteaga-Cedeño, Miguel Á. Carbonero-Martín, Luis J. Martín-Antón, Paula Molinero-González

**Affiliations:** Grupo de Investigación de Excelencia GIE-179, Departamento de Psicología, Facultad de Educación, Universidad de Valladolid, Paseo Belén N 1, 47011 Valladolid, Spain

**Keywords:** emotional intelligence, primary school teacher, infant school teacher, career profile, sociodemographic profile

## Abstract

Emotional intelligence is the key to students’ psychological-social well-being and academic performance, and teachers must provide socioemotional education in the classroom. To achieve this, teachers must display high levels of socioemotional skills that ensure their own personal, social, and career well-being and, as a result, that of their students. This study seeks to gain an insight into the levels of socioemotional skills of emotional perception, emotional understanding, and emotional regulation and how these are linked to the sociodemographic and career profile of teachers in infant and primary education. For this, we surveyed 351 teachers—310 female and 41 male—aged between 20 and 69. We used the Trait Meta-Mood Scale-24 (TMMS-24) together with a questionnaire (ad hoc) to determine the sociodemographic and career profile of participating teachers. Statistical analysis of the data showed that gender impacts on emotional regulation and emotional perception, while age and number of the children were also seen to have significant difference on emotional regulation and emotional perception. In contrast, professional qualifications were only seen to affect emotional perception. The variable reflecting the level at which staff teach showed significant difference on emotional perception and emotional understanding, while administrative posts held by teachers also demonstrated significant difference on their emotional understanding. The results confirm that sociodemographic and work-related variables impact the level of socioemotional skills of infant and primary education teachers. These aspects should be taken into account in the structure and planning of training aimed at developing socioemotional skills in order to ensure their success.

## 1. Introduction

Emotional intelligence represents the ability to perceive, use, express, and regulate one’s own emotions as well as those of others by processing emotional information [1]. Socioemotional skills are a key personal resource vis-à-vis perceiving, assimilating, and regulating moods [2]. Emotional intelligence enables the individual to adapt, take on challenges, increase their positive thoughts and increase their chances of success [3]. Emotional intelligence predicts good mental and physical health [4]. Emotional intelligence allow individuals to face up to and deal with the complex, stressful, and shifting situations encountered in today’s world [5]. Given the key contribution that socioemotional skills can make to people’s well-being and success, developing such skills in the education system has become of ever-increasing importance over the last few years.

Emotional intelligence is currently one of the best predictors of success in the academic and professional field [6,7,8]. Training in socioemotional skills holds tremendous potential in terms of boosting personal, career, academic, and social well-being amongst teachers [9]. Emotional intelligence ensure that teaching staff are able to adopt the tools, skills, attitudes, and behaviors that stimulate optimistic thoughts, exploration of emotions, conflict-solving, and an appropriate teaching-learning process [3,10,11]. The inclusion of socioemotional education in the curriculum at different levels of training has been proposed by various international bodies as well as by a number of authors in recent years [9,12].

The Organization for Economic Cooperation and Development (OCDE) [13] included socioemotional skills in its evaluation of skills due to their providing the basis for students’ psychological-social well-being and academic success. UNESCO [14] sets out that socioemotional skills should be worked on similar to any other school subject. Socioemotional education should be implemented in the classroom by teaching staff themselves, since more effective results are achieved than with programs or activities led by non-academic staff [15,16,17]. Teaching staff should therefore be the first to receive socioemotional education [18,19,20,21], since this would provide them with an effective model for handling socioemotional skills when they are with their students [22,23,24,25].

Developing socioemotional skills contributes significantly towards the overall well-being of teachers [26], with socioemotional well-being proving key if teachers are to educate effectively [19,27]. Emotionally intelligent teachers enhance their students’ intrapersonal and interpersonal relations [7,28] and motivate them to cope effectively with the conflictive and stressing situations that emerge in schools [29], in addition to helping students achieve enhanced academic performance [21]. Given the benefits of socioemotional skills amongst teaching staff and their importance for students, training in socioemotional skills should be deemed a primary objective in both the initial as well as ongoing training of teaching staff [30,31].

Socioemotional skills should be implemented in lifelong training through intervention programs, which are considered the best educational orientation strategy [19,32,33,34,35,36]. Well-grounded, well-planned, and structured intervention programs in emotional intelligence [19] that set out clear and specific objectives [37], and which adopt a theoretical, practical as well as experience-focused approach will yield superior results in the long term [3,38,39]. Socioemotional skills should also be included in the curricula of teaching staff’s initial training and should offer specific content geared towards developing such skills [17,38,40,41]. Implementing socioemotional education in teachers’ initial and ongoing training will enable them to recognize their limitations and to deal effectively with the challenges posed by their profession [29,42].

Teaching is one of the professions that involves the greatest emotional charge [43,44]. Teaching staff engage in intense personal interactions with students, parents, and workmates [45,46,47], and are subject to excessive workloads and demands. Moreover, they often lack sufficient social support and must address the management difficulties they are faced with every day in the classroom [48,49]. These factors may lead to teacher dissatisfaction with their work [50], and thereby limit their professional performance and personal well-being [40,51,52].

For teachers, achieving success in personal, social, and work-related aspects involves developing socioemotional skills that allow them to control their emotional states when faced with the various situations they encounter [1,2,53,54,55]. Skills such as emotional perception, emotional understanding, and emotional regulation enable them to use the information afforded by their own emotions in order to improve their thinking in the various spheres of socialization [54,56]. Teachers who are able to develop emotional perception to an intermediate level (paying neither too much nor too little attention to emotions and feelings) have the ability to perceive, value, and express emotions with precision [21]. Educators who are able to develop emotional understanding display the capacity to comprehend and appreciate the emotional significance of the situations they experience [1]. Teachers who exhibit high expectations in terms of emotional regulation have a greater chance of using coping strategies [24] and display greater psychological adjustment and emotional well-being [57]. It is clear that the enormous benefits provided by developing socioemotional skills have today enabled key research to be carried out into the field.

Some studies have explored what impact certain sociodemographic and work variables might have on the development of socioemotional skills amongst teaching staff [2,7,9]. Gender and age are two of the sociodemographic variables influencing the development of socioemotional skills that have been explored in various studies. One study highlights that women display higher levels in socioemotional skills [9], whereas another study shows that it is men who achieve higher levels [58]. Women show greater levels in understanding and emotional regulation [2], while men tend to score higher in emotional regulation [31,59,60]. Other studies posit that there are no significant differences between gender and socioemotional skills [26,61]. With regards to age, some studies find that socioemotional skills improve with age [9,62], whereas others report that the higher the age the lower the emotional regulation [63] and the less interest there is in perceiving emotions [64]. Another study shows that there are no statistically significant differences between age and socioemotional skills [65].

Amongst work-related variables, teachers are found to display less interest in emotional understanding as the number of years spent in the profession increases [64], while work experience has been seen to enhance the levels of socioemotional skills when addressed during the process [66]. It has also been reported that work experience has no impact on socioemotional skills [2]. The geographical area in which teachers work has also been reported to exert an influence on socioemotional skills. Rural areas have been to seen to favour emotional perception and emotional regulation whilst urban areas favour emotional understanding [7]. Some findings offer results which show that students in urban areas display higher levels of socioemotional skills compared to students in rural schools [67]. It has also been reported that the teacher’s specialty (infant and primary) as well as any administrative posts that teachers might have held have no influence on the development of socioemotional skills [2,21].

The results reported in the above-mentioned studies are varied. Sociodemographic variables such as gender and age have been highlighted, yet it is not known whether teachers’ civil status, level of qualification and number of their own children influence the development of socioemotional skills. Different results have thus emerged concerning work-related variables such as years of experience, the geographical area where the teachers’ school is located, and the administrative posts they might have held. Nevertheless, we believe that it is important to determine whether teachers’ level of training, work dependency, and pay grade might impact the levels of socioemotional skills they achieve. Despite this, no results on these topics have been found in the literature reviewed. Based on all of the above information, this study seeks to gain an insight into the levels of socioemotional skills of emotional perception, emotional understanding, and emotional regulation and their link to the sociodemographic and career profile of infant and primary education teachers.

## 2. Materials and Methods

### 2.1. Participants

This research considered teachers in infant education (initial and preparatory) and primary education (elemental and mid) from the district of El Carmen, in the province of Manabí, Ecuador, where there are a total of 1317 teachers. Of this total, 27% took part in our study. The sample was made up of 351 infant and primary education teachers from public schools (11 in the urban area and eight in the rural area) in the city of El Carmen, in the province of Manabí, Ecuador. As can be seen in Table 1, the sociodemographic profile shows a clear distribution bias towards women. As regards age, the most representative age ranges were 30–39 and 40–49 years old. Most of the teachers in the research sample were married. As regards the level of qualifications reached, worthy of note is the fact that a significant percentage of teachers only have a short degree qualification.

Analysis of the socio-career profile of the teachers in our study—as can be seen in Table 2—shows that most of those in our sample correspond to primary education. In terms of contract situation, most of the teachers have obtained a definitive appointment, and most work in urban areas. It is worth highlighting that in the teaching scale category –which reflects pay grade– teachers in our study are in the lowest level categories. In other words, the categories of teaching staff who have recently gained appointments, whereas the higher-level categories contain only quite a small number (teachers can be promoted when they fulfil certain requirements such as years of work experience, participation in induction and/or mentoring courses, a certain number of hours’ participation in ongoing training courses, and obtaining a specific mark in teaching assessment). Also worth mentioning is that a high percentage of our sample have between 10 and 20 years of teaching experience. Finally, most of our teachers have not held administrative posts in education.

### 2.2. Instruments

The Fernández-Berrocal et al. [68] Trait Meta-Mood Scale-24 (TMMS-24) was used, with a Spanish adaptation of the original Salovey et al. [69] questionnaire. The TMMS-24 scale is a self-report test that measures emotional intelligence. It is structured in 24 items measured on a 5-point Likert scale (1 = totally disagree, and 5 = totally agree). The TMMS-24 scale evaluates three dimensions, with 8 items in each category: emotional perception (level of belief in one’s emotional focus); emotional understanding (subjective perception of one’s own emotions), and emotional regulation (belief in the ability to curb and regulate negative emotional moods and to boost positive ones). In our case, the internal reliability reached in the three dimensions was 0.87 (perception), 0.87 (understanding), and 0.81 (regulation), which are very similar data to those presented in previous studies, in which internal reliability varied between 0.84 and 0.88 [64,68,70,71].

A questionnaire was drawn up to determine the sociodemographic profile of teachers, taking into account: gender, age, civil status, and qualifications. Information was also gathered regarding career profile aspects such as contract situation, level taught, area where the school is located, the teaching scale category (pay grade), years of work experience, and having held an administrative post.

### 2.3. Procedure

The study was approved by the ethics committee of the authors’ home university. The research complied with Ecuador’s Organic Law Governing Protection of Personal Details. Subsequent to having received approval, the district directors in charge of organizing and overseeing educational, administrative and institutional management issues were contacted in order to gain their authorization to enter the schools. Permission was quickly granted by the head teachers and independent meetings were held with each school, wherein we explained the objectives and method and described the research tools, in addition to making it clear that any details would be treated anonymously and confidentially. The data collection method, which involved filling in the questionnaires through Google Forms, was explained and participants were provided with a single link that contained the two instruments used in the study. This form contained an explanatory text describing each instrument together with an informed consent form so as to ensure that teachers were fully aware of the aims of our research and of the voluntary nature of their participation.

### 2.4. Data Analysis

All of the statistical analyses were performed with the help of R Studio v 1.4.1106 software. Given that the data did not present a normal distribution, non-parametric techniques were used to compare means between groups. In order to compare the means of the two groups, we used the Wilcoxon signed rank test, and Cohen’s d was used to quantify the size of the effect of the factor studied. The Kruskal-Wallis test was used to compare the means of three or more groups.

## 3. Results

### Descriptive Analysis of Emotional Intelligence

As regards the results obtained for factors concerning socioemotional skills, Table 3 shows how those which are most developed by infant and primary school teachers are emotional regulation (M = 34.93), followed by emotional understanding (M = 33.95), and then emotional perception (M = 29.72).

As regards the Pearson correlation—as can be seen in Table 4—the summary variables (i.e., the three factors that make up the model) show a correlation that is between low and medium.

Table 5 shows how women score higher than men in the emotional perception factor, whereas men achieve higher levels in emotional regulation when compared to their female counterparts. Teachers’ age shows that as they grow older their levels of emotional perception decline. The civil status variable has no statistically significant difference on any of the factors involved in emotional intelligence. Finally, the number of their own children that teachers have is seen to exert a significant difference on the emotional perception and emotional regulation factor. The lower the number of children the higher the level of emotional perception, and the greater the number of children the higher the levels of emotional regulation.

Table 6 shows the results of the variables considered in teachers’ career profile. The variable “professional qualifications” is seen to achieve significant results in the emotional perception factor, whereas teachers who lack a professional qualification related to teaching had lower levels of emotional perception than teachers who obtained a short degree and a master’s degree and who show higher levels of emotional perception. The variable “contract situation” has no significant difference on any of the socioemotional skill factors. The variable “level taught” displays a significant relationship with the emotional perception and emotional understanding factor, with infant education teachers achieving higher levels in the two factors when compared to primary education teachers. The variable “area where the school at which the teacher teaches is located” shows no significant differences in any of the socioemotional skill factors. The variables “teaching scale category” (pay grade) and “years of teaching experience” show that there are no significant differences in any of the emotional intelligence factors. Finally, teachers who do not hold or who have not held any administrative post are seen to display better levels in the emotional understanding factor.

## 4. Discussion

The factor displaying the highest levels of emotional intelligence amongst the infant and primary education teachers in our study is emotional regulation, followed by emotional understanding and, finally, emotional perception. These data are similar to those published by Cazalla-Luna and Molero [31], Domínguez and Nieto [7] and Schoeps et al. [43], Martínez-Saura et al. [21] found that infant and primary education teachers exhibited high levels of emotional perception, then emotional regulation and, finally, emotional understanding. Suárez and Martín [2] reported that teachers in higher education obtained more significant levels in emotional understanding, then in emotional regulation and, finally, in emotional perception. The results to emerge from the research into the levels of development of emotional intelligence factors clearly differ, which might be due to the educational levels which the various studies have focused on.

It is worth highlighting that emotional intelligence factors (emotional perception, emotional understanding, and emotional regulation) should be developed in an integrated manner. No single factor is more important than any other and all of them together exert a significant difference in terms of achieving individuals’ overall well-being [26,72,73]. However, when evaluating levels of emotional intelligence, different authors agree that it is important to develop the highest possible levels of emotional regulation and understanding in teachers. As a result, moderate levels of emotional perception should also be sought since this factor can foster aspects of adaptation in which teachers are able to evaluate events and respond appropriately and correctly to them. It is also essential to develop adaptive aspects in order to avoid situations in which teachers understand what they are feeling, yet respond inappropriately [1,21,31].

In our study, the three factors that make up the model display a correlation between low and medium. These results are very similar to those published in the research conducted by Valente and Lourenço [25]. Some studies contend that the greater the emotional understanding the greater the emotional regulation and, therefore, the greater the satisfaction in terms of life and well-being [74]. Moderate levels of emotional perception correlate positively with emotional understanding and regulation [41]. Another study finds that greater emotional regulation is positively linked to emotional understanding and emotional perception and, therefore, with better coping and conflict-solving strategies [75]. The correlation between the different emotional skills factors confirms what was mentioned previously regarding how essential emotional intelligence are for teacher interaction with all of the stakeholders involved in the educational system. Moreover, being able to jointly develop the factors that make up emotional intelligence boosts the general and comprehensive well-being of the individual.

As regards the impact of the teachers’ socio-demographic profile on emotional intelligence, we noted that gender has a significant difference on two of the factors that make up emotional intelligence. Women tend to display greater levels of emotional perception, whereas men achieve higher levels in emotional regulation, with a number of studies supporting our results [7,31,59,60,64,76]. Other studies have underpinned how gender impacts other factors, such as Suárez and Martín [2], who maintained that women exhibit better levels in emotional understanding and regulation. Pena et al. [77] stated that women only score higher in the perception factor, whereas men obtain better levels of emotional understanding and emotional regulation. We have also found several studies stating that there are no significant differences with regards to gender and emotional intelligence [26,61,78,79].

The age variable has a significant effect on levels of emotional perception. The lower the age the greater the emotional perception, whereas the greater the age the lower the levels of emotional perception. These data concur with those published by García-Domingo [64] and Domínguez and Nieto [7] who stated that as people grow older they tend to pay less attention to and reflect less on their emotions and moods. Some authors also mentioned that the greater the age the lower the emotional regulation [63], while other authors have shown that emotional intelligence improve with age [9,62,79]. There are also studies highlighting that no statistically significant differences exist with regards to age and emotional intelligence [2,65,80].

Teachers need to be provided with activities that are well-planned and structured and that set out clear and specific objectives so as to be able to develop the right levels in emotional perception, understanding, and regulation, independent of gender. Also to be taken into account are teacher age and the degree to which they have been able to develop each factor (emotional perception, emotional understanding, and emotional regulation) so that they can acquire a knowledge of strategies that will enable them to strike the right balance of emotional perception and which will help them boost the regulation and understanding of their emotions.

Another sociodemographic variable that evidences a significant difference in our study is the number of children that teachers have to manage. This variable has an impact on the development of emotional intelligence, particularly in terms of emotional perception and regulation. The greater the number of children, the lower the levels of emotional perception, and the greater the number of children, the greater the levels of emotional regulation. However, we have found few results with which to compare our findings. Salinas-González [81] carried out a study with university teachers and contend that the number of children has no bearing on the level of emotional intelligence, whereas Morand’s [82] study of how family size impacts emotional intelligence posits a positive correlation between the two variables. In their study of parents and emotional intelligence, Platsidou and Tsirogiannidou [83] stated that the number of children has no significant difference on parents’ emotional intelligence levels. Given the limited amount of information available, we believe that it is important to take these variables into account when conducting future studies so as to establish possible linkages between the number of children in the family and the various factors that make up emotional intelligence.

With regards to higher qualifications obtained by teachers, results show that those who have not been trained as teachers and do teach tend to evidence lower levels of emotional perception, whereas those with degree or a master’s degree display higher levels of emotional perception. This could be attributed to the fact that teachers in this case are to some extent unfamiliar with the duties they must perform and tend to pay less attention to certain details that are key to those who have been trained as teachers. However, we have found no studies that have examined this issue, such that we highlight the importance of future research exploring what impact teachers’ qualifications might have on the development of their emotional intelligence.

As it concerns the teachers’ career profile it was found that the variables which showed significant difference on the development of emotional intelligence in our sample were the level at which teachers taught and the administrative posts they hold or have held. Amongst the infant and primary education groups, it is worthy of note that infant education teachers—who are mostly women (99.05%)—display higher levels of overall emotional intelligence and achieve significant scores in emotional perception and understanding. Our results concur with those published by Cejudo et al. [84] who reported that infant education teachers present better levels in emotional intelligence compared to their counterparts in primary education. Escolar et al. [85] noted that infant/preparatory education teachers show greater interests in ongoing training and innovation on issues related to emotional intelligence. The significantly high levels of emotional intelligence achieved by infant education teachers might be a result of the fact that at these levels, the lessons take place in more playful and interactive atmospheres [86].

The fact that infant/preparatory education teachers in our sample are “open” to training on emotional intelligence might be due to the fact that the curriculum in the Ecuadorian educational system for this level of teaching specifically contains the area of socioemotional development [87]. Early school teachers consider that the development of emotional intelligence is key to performing their educational duties [61]. Nevertheless, our results do contrast with those reported by Cazalla-Luna and Molero [31] and García-Domingo [64]. Said authors stated that primary education teachers display higher levels of emotional intelligence compared to infant school teachers. García-Tudela and Marín-Sánchez [65] contend that both infant education and primary education involve play activities such as gamification and game-based learning, amongst other strategies, that enable work to be carried out with issues linked to emotional intelligence, and which can directly impact teachers’ emotional intelligence. Some authors claim that the educational stage at which teachers teach has no significant difference on the levels of emotional intelligence developed [21].

The results show that teachers who do not hold or have not held an administrative post exhibit high levels in emotional understanding compared to teachers who have held such posts. Our findings contrast with those reported by Castellano [88] who mentioned that university teachers who have held administrative/management positions present higher levels of emotional intelligence. In their study carried out on university teachers, Suárez and Martín [2] maintained that administrative duties undertaken by teaching staff have no bearing on the levels of emotional intelligence. It could be argued that our results are due to the fact that teachers who give up the classroom for some time in order to take up an administrative post tend to disconnect from teaching and from the work they have to do in the classroom and that returning to teaching proves difficult in terms of resuming their teaching duties. As we were unable to find any further information concerning how these sociodemographic and career-related variables impact infant and primary education teachers, we urge future research to explore what significant differences these variables might have on emotional intelligence. We also suggest differential analysis be carried out on teachers (that is, those who only undertake administrative tasks for a certain period and those who perform both tasks at the same time, in addition to taking into account the remaining variables).

Sociodemographic and career variables have been insufficiently addressed in research that has focused on the development of emotional intelligence in teachers [2,7,89] Gabel-Shemueli et al. [90]. Sociodemographic and career variables are relevant, but they are not always determining factors and tend to increase people’s vulnerability to the emotions they experience [90,91]. Personal and career variables tend to be essential in the way teachers perceive, interpret and respond to the emotions experienced in the different areas of socialization [16,18,92,93]. It is important that teachers develop and/or strengthen emotional intelligence to a level that is linked to the known impact of those sociodemographic and career variables that have been identified with significant differences in relation to the levels of global emotional intelligence and in each one of the factors that integrate it (emotional perception, understanding and regulation [1].

Significant differences were found in our study on the factors of emotional intelligence between the groups of teachers that are divided based on gender, age, number of children, professional degree, and the level at which the infant teachers do their work and primary in relation to the emotional perception factor. Other variables that, when considered, allow the creation of groups of teachers (among whom there are significant differences with respect to emotional intelligence factors) are the level at which they teach and the administrative management carried out by the infant and primary education teachers in relation to emotional understanding. Finally, the variables that showed significant differences in relation to emotional regulation were gender and number of children.

It is important to note that teachers who achieve moderate levels of emotional perception, identify and recognize emotions and feelings in themselves and in others, identify emotional expression (gestures, body movements, and tone of voice), the physiological and cognitive sensations that own emotions activate [34,94]. Teachers with high levels of emotional understanding identify emotional expressions, name the emotions, apply anticipation and retrospection strategies to identify the possible causes that generate moods and the consequences of their actions [95,96]. Emotional regulation is one of the most complex skills to develop. Teachers with high levels of emotional regulation are prone to positive and negative feelings, capable of processing information and promoting an understanding of personal growth, and applying strategies to regulate their emotions and the emotions of others [1,65].

Teachers with high levels of emotional intelligence manage to integrate emotional experiences with thoughts and actions [97]. Education professionals with high levels of emotional intelligence are able to adequately manage their own personal, work, academic and social well-being [98,99]. Teachers with high levels of emotional intelligence are capable of providing psychological well-being to their students and also provide them with strategies that allow them to handle events in which emotions are involved [56]. The development of emotional intelligence in teachers is key to maintaining a positive and pro-social classroom climate and promoting the academic success of students [17]. The emotional intelligence of teachers must be addressed through the implementation of emotional education programs that take into account the sociodemographic and career variables of the participants. [23,24,43].

## 5. Conclusions

The evidence from this study highlights how vital it is to take into account teachers’ sociodemographic and career profile background so as to use such situations as a basis for designing resources based on the characteristics and needs of those teachers who will be given training. From this perspective, we propose that, as suggested by other authors [2,9,25], priority should be given to teachers’ personal characteristics (sociodemographic and career profile) when devising educational programs aimed at developing emotional intelligence.

In the same vein, we propose that the implementation of interventions be developed in a segmented manner based on variables of the sociodemographic profile (i.e., age, gender, number of children that the teaching staff has, and the academic degree of the professionals dedicated to training). Another type of organization that we propose for the interventions is to carry out segmentations considering the career profile of the teaching staff (i.e., the level to and way in which the teaching staff teaches, the geographical location of the educational institution where the teaching staff works, and the administrative management carried out by the teaching staff). We also propose that segmentations can be considered where the sociodemographic and career profile of the teaching staff is integrated, contemplating (i.e., the age and level of training in which they are teaching classes or segmentations considering gender with the studies achieved by the teaching staff).

Another proposal is the implementation of programs focused on teachers to consider the levels attained in general emotional intelligence and in each of its aforementioned factors. Segmenting the work teams while considering the different factors of emotional intelligence (emotional perception, understanding and regulation) in which the teachers have shown significant differences in their previous evaluations (pre-tests). This will highlight and create room for strengthening whatever weaknesses that some teachers might demonstrate. It is important that teachers do not know details of how the segmentations are organized to avoid difficulties in interaction.

The implementation of the programs through the organization of segmentations will create room for attending to teachers who have similarities in the sociodemographic and career variables and in the factors that make up emotional intelligence. The strategies and activities that are proposed depend on the characteristics and needs of each segment. The implementation of intervention programs is also proposed for the development of emotional intelligence in teachers and strategically structuring the application through the different segments that are considered and, at the same time allow teachers to strengthen their emotional intelligence through interactive, experiential and cooperative learning.

It is necessary to carry out a permanent evaluation of the program sessions to identify if the organization with the selected segmentations are appropriate for the purpose. We also believe that it is important that, if the implementation of the programs has a large number of participants, the different sessions can be held by reorganizing the teams and taking into account other types of segmentation according to the needs that are deemed appropriate. These ways of organizing the proposed interventions that can be developed with the teaching staff at the different levels of training will allow the different factors that make up emotional intelligence to be strengthened in a personalized way.

This study has demonstrated which sociodemographic and career variables show significant differences in relation to global emotional intelligence and each of its factors. The issues of gender, age, number of children, professional degree and the level at which infant and primary education teachers teach their classes have highlighted and show significant differences with the emotional perception factor. With regards to these variables, level at which they teach and the administrative management carried out by the infant and primary education teachers show significant differences with emotional understanding and, finally, the variables that showed significant differences in relation to emotional regulation were gender and number of children. This finding is important since it permitted the identification of the need to consider the sociodemographic and career profile for the structure of future intervention programs aimed at teachers. It has permitted the suggestion of strategies to achieve an adequate implementation by organizing the teams of participating teachers through the different forms of segmentation. It further allows for an emphasis to be placed on the need to carry out the initial evaluation to detect the needs of the participants and adapt appropriate strategies and activities for each session. In the same way, this work highlights the importance of permanent evaluation to attend, in a precise and concrete way, to the needs that arise on the teaching staff during the development of the program. Finally, it contemplates the need to carry out the final evaluation and analysis to determine the efficiency, effectiveness, and level of impact of the program on the participating teachers.

Given the findings presented in this study, it is recommended that future research works examine a comparative analysis at the international level between sociodemographic and career variables in relation to emotional intelligence, considering education professionals as a study group in order to establish a contrast with the presented results. We also consider it necessary to design an intervention program in emotional education while considering sociodemographic and career profiles and implement it with the group of teachers who have participated in the present study to determine if the sociodemographic and career variables show significant differences in the teaching staff post intervention while applying the proposed segmentation strategies.

## Figures and Tables

**Table 1 ijerph-19-09882-t001:** Sociodemographic profile of the sample of infant and primary education teachers.

Characteristics	Frequency (n = 351)	%
Gender:		
Male	41	11.7%
Female	310	88.3%
Age: (m = 44.07)		
20–29	12	3.4%
30–39	112	31.9%
40–49	118	33.6%
50–59	94	26.8%
60–69	15	4.3%
Civil status:		
Single	26	7.4%
Single with children	99	28.2%
Married	169	48.1%
Widowed	13	3.7%
Divorced	44	12.5%
Number of children:		
No children	37	10.5%
1 child	60	17.1%
2 children	120	34.2%
3 children	100	28.5%
4 children	26	7.4%
5 children	8	2.3%
Professional qualification:		
Teaching degree	19	5.4%
Full degree	266	75.8%
Short degree	5	1.4%
Master’s degree	53	15.1%
Qualifications not related to teaching	8	2.3%

**Table 2 ijerph-19-09882-t002:** Career profile of the sample of infant and primary education teachers.

Characteristics	Frequency (n = 351)	%
Level taught:		
Initial/Preparatory	106 (M = 1 F = 105)	30.2%
Basic Elemental/Mid	245 (M = 40 F = 205)	69.8%
Contract situation:		
Temporary contract	31	8.8%
Provisional appointment	25	7.1%
Permanent appointment	295	84.0%
Geographical area:		
Urban	252	71.8%
Rural	99	28.2%
Teaching scale category:		
F-G	304	86.6%
D-E	31	8.8%
A-B-C	16	4.6%
Years of teaching experience: (m = 14.82)		
Less than 10	105	29.8%
11 to 20	175	49.9%
21 to 30	56	16.0%
Over 30	15	4.3%
Administrative posts held:		
Yes	69	19.7%
No	282	80.3%

**Table 3 ijerph-19-09882-t003:** Description of items by the factors that make up emotional intelligence.

**Perception (M: 29.72; SD: 6.42)**	**M**	**SD**
I pay a great deal of attention to feelings.	4.37	0.78
I am normally very concerned about what I feel.	4.10	0.98
I usually spend time thinking about my emotions.	3.91	1.05
I think it is worth paying attention to my emotions and moods.	4.33	0.88
I allow my feelings to affect my thoughts.	2.39	1.25
I am constantly thinking about my mood.	3.24	1.33
I often think about my feelings.	3.52	1.27
I play a great deal of attention to how I feel.	3.83	1.15
**Understanding (M: 33.95; SD: 4.95)**	**M**	**SD**
I am clear about my feelings	4.54	0.70
I am often able to define my feelings.	4.35	0.75
I almost always know how I feel.	4.43	0.71
I usually know what I feel about people.	4.10	0.92
I often realize what my feelings are in different situations.	4.19	0.81
I am always able to say how I feel.	4.09	0.99
I am sometimes able to say what my emotions are.	3.97	0.97
I am able to understand my feelings.	4.24	0.82
**Regulation (M: 34.93; SD: 4.64)**	**M**	**SD**
Even though I do at times feel sad, I tend to have an optimistic outlook.	4.46	0.76
Even when I feel bad, I try to think about pleasant things.	4.51	0.74
When I am sad, I think about all of the pleasures life has to offer.	3.97	1.15
I try to have positive thoughts, even when I feel bad.	4.52	0.72
If I think too much about things, complicating them, I try to calm down.	4.18	0.98
I try to make sure I am in a good mood.	4.36	0.94
I have a lot of energy when I feel happy.	4.73	0.58
When I am angry I try to change my mood.	4.17	0.96

Note. M: mean; SD: standard deviation.

**Table 4 ijerph-19-09882-t004:** Correlation matrix between the factors that make up emotional intelligence.

Factors	Emotional Perception	Emotional Understanding	Emotional Regulation
Emotional perception	1.00	0.33	0.25
Emotional understanding	0.33	1.00	0.57
Emotional regulation	0.25	0.57	1.00

**Table 5 ijerph-19-09882-t005:** Significant difference of the variables that make up the sociodemographic profile in emotional intelligence.

Factors	Emotional Intelligence		
	Emotional Perception	Emotional Understanding	Emotional Regulation
Gender			
Male	28.68 (3.39)	33.78 (4.67)	34.24 (3.85)
Female	37.86 (2.77)	33.98 (5.91)	28.02 (4.87)
*p*-value	0.016 **	0.95	0.042 **
Cohen’s d (r)	2.96	−0.04	1.42
Age			
Up to 29	33.41 (4.23)	33.52 (5.09)	37.11 (2.23)
30–39	30.87 (5.92)	34.39 (4.49)	35.02 (4.32)
40–49	29.29 (6.71)	33.27 (5.24)	35.31 (4.17)
50–59	28.86 (6.61)	34.40 (5.11)	34.40 (5.71)
60 or over	27.06 (6.08)	33.28 (4.82)	33.49 (4.14)
*p*-value	0.015 **	0.33	0.27
Number of children			
No children	33.11 (2.87)	29.11 (3.99)	25.13 (3.11)
1 child	31.67 (3.01)	29.34 (5.12)	26.42 (4.12)
2 children	29.44 (4.14)	29.23 (2.09)	26.11 (2.87)
3 children	28.11 (3.67)	27.078 (3.56)	27.98 (3.14)
4 children	27.06 (5.17)	29.09 (6.32)	34.56 (5.10)
5 children	25.78 (2.95)	30 (4.12)	36.98 (4.12)
*p*-value	0.016 **	0.14	0.021 **
Professional qualification			
Teaching degree	28.4 (4.32)	35.4 (3.53)	34.5 (5.65)
Full degree	29.7 (6.55)	33.8 (5.14)	35.0 (4.58)
Short degree	31.2 (6.46)	34.6 (4.28)	33.0 (3.87)
Master’s degree	30.9 (6.14)	34.4 (4.10)	34.6 (4.57)
Qualifications not related to teaching	24.0 (5.98)	32.0 (6.89)	35.8 (5.87)
*p*-value	0.03 **	0.78	0.56

Note. ** The difference is significant at 0.05 level.

**Table 6 ijerph-19-09882-t006:** Significant difference of the variables that make up the career profile in emotional intelligence.

Factors	Emotional Intelligence		
	Emotional Perception	Emotional Understanding	Emotional Regulation
Contract situation			
Temporary contract	28.1 (6.51)	33.7 (4.94)	35.5 (3.65)
Provisional appointment	27.8 (8.14)	34.0 (6.07)	36.0 (4.41)
Definitive appointment	30.1 (6.22)	34.0 (4.87)	34.8 (4.75)
*p*-value	0.15	0.79	0.33
Level			
Initial education	30.6 (6.32)	34.5 (4.42)	34.9 (4.02)
Primary education	25.1 (2.14)	27 (5.15)	34.9 (4.90)
*p*-value	0.031 **	0.041 **	0.46
Cohen’s d (r)	1.08	0.97	0.0065
Area where the teacher’s school is located			
Urban	30.0 (6.39)	33.9 (4.90)	34.8 (4.86)
Rural	28.9 (6.48)	34.0 (5.01)	35.3 (4.10)
*p*-value	0.17	0.79	0.66
Cohen’s d (r)	0.16	−0.017	−0.10
Category of teaching scale (pay grade)			
F-G	34.00 (4.43)	30.28 (3.37)	30.19 (6.27)
D-E	33.32 (5.83)	30.00 (5.21)	34.09 (4.94)
A-B-C	34.06 (3.56)	29.00 (6.39)	33.19 (4.79)
*p*-value	0.94	0.63	0.184
Years of work experience			
Less than 10 years	29.3 (6.58)	33.7 (4.84)	35.2 (3.99)
11 to 20 years	30.2 (6.47)	34.3 (4.89)	35.1 (4.57)
21 to 30 years	29.6 (6.06)	33.2 (5.51)	34.0 (5.55)
Over 30 years	27.9 (6.18)	35.3 (4.10)	34.8 (6.12)
*p*-value	0.37	0.35	0.60
Administrative post held			
Yes	29.5 (6.14)	27.1 (5.08)	34.6 (4.97)
No	29.8 (6.50)	34.1 (4.90)	35.0 (4.57)
*p*-value	0.70	0.039 **	0.69
Cohen’s d (r)	−0.049	−1.01	−0.074

Note. ** The difference is significant at 0.05 level.

## Data Availability

The data presented in this study are available subsequent to a request submitted to the corresponding authors.

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
