# Peer review of "The Sociodemographic-Professional Profile and Emotional Intelligence in Infant and Primary Education Teachers"

_ijerph, 2022, doi:10.3390/ijerph19169882_

Round 1

Reviewer 1 Report

The general approach of the study is not appropriate since it is stated in terms of socio-emotional skills and the test used (TMMS-24) is an instrument that assesses trait emotional intelligence. Therefore, it is recommended that the title be modified, as well as the wording of the results and discussion using a terminology more in line with this other construct.

The discussion is written using expressions such as "influence" or "effect", relating them to the sociodemographic variables studied, when the research approach has been descriptive. Therefore, drawing conclusions in these terms is inappropriate. We recommend that other statistical analyzes of a predictive type be carried out or that the writing of the conclusions be oriented towards verifying the differences between the groups compared, based on the sociodemographic variables.

Reviewer 2 Report

The article presented shows evidence of how the sociodemographic and career profile influences the development of socio-emotional skills in teachers of early childhood and primary education. I consider relevant to highlight the following aspects:

1) The contextual and theoretical framework of the state of the art is complete and relevant.

2) The theme of work is current, necessary and transferable to realities of different educational levels

3) The methodology and research techniques used are validated, endorsed and show rigor in the procedure

4) It shows us a panorama in which we have areas for improvement that are easily implementable in reality.

As areas for improvement, the sample could be expanded by increasing international references and making proposals for implementation in reference centers.

Reviewer 3 Report

Thank you to the author(s) for their work on this paper. This is an interesting area and I was grateful for the opportunity to review and learn about their research. I hope the following feedback is helpful in further developing the paper.

Overall the paper is strong, with the background addressed informatively and insightfully. It was good to see ethical considerations addressed in section 2.3 - thank you for including this. 

Findings are clearly presented though at times there are areas which would benefit from a more detailed write-up. 

Re: line 304, the reference to women and men could perhaps be adjusted to "regardless of sex or gender" or "across genders" or similar to avoid a rigid binary approach.

Re: lines 377 - 380, this section would benefit from more critical discussion. To argue that teachers should be the source of their wellbeing and the wellbeing of others ignores a lot of structural and systemic issues which are hazards to wellbeing, individually and collectively. I recommend that this section be fleshed out with more depth, detail, and critical insights.

Re: lines 392 - 402, this is an interesting recommendation but I think in need of further development. A single paragraph on this feels like a beginning point rather than a conclusion. What I would suggest is that these ideas be developed across a couple of paragraphs so that readers gain full insight, then wrap up the paper with a final concluding paragraph which draws everything together. 

Some minor copy-editing is also needed across the paper. 

Round 2

Reviewer 1 Report

Although the researchers have partially taken into consideration the indications suggested, I must warn that the focus of the work is exclusively descriptive. This reduces the level of relevance of the results obtained, as well as the originality of the conclusions drawn from them.

There is already a multitude of investigations that descriptively study social and academic variables of teachers, so this work only offers a particular contextualization of what is already known.

We consider that, if its methodological approach had been more predictive, using some type of multivariable analysis, taking advantage of the continuous nature of some of the variables studied (age, years of experience, etc...) it would be a sufficiently relevant contribution to provide scientific results. that expand the body of research on emotional intelligence.

Therefore, given that this assessment refers to the general relevance of the study, I consider that the editorial managers should be the ones who make the decision on its publication.
